# Oceanic crust recycling controlled by weakening at slab edges

Jessica Munch [1✉], Taras Gerya [1] & Kosuke Ueda[1]

Retreating subduction zones such as the Lesser Antilles, Gibraltar and Scotia have been migrating towards the Atlantic Ocean by cutting their way through the oceanic crust. This spontaneously retreating subduction is enabled by the development of faults at the edges of the slab, but the physical mechanisms controlling fault propagation and direction remain unknown. Here, using 3D numerical subduction models we show that oceanic lithosphere recycling is mainly controlled by the intensity of strain-induced weakening of fractures forming at the edges of the slab. Intense strain-induced weakening causes predominantly brittle fault propagation and slab narrowing until detachment. Without weakening, preponderantly ductile slab edge propagation occurs, which causes slab widening. This rheological control is not affected by the proximity of non-weakened passive continental margins. Natural examples suggest that slab edges follow convergent paths that could be controlled by fractures weakening due to deep water penetration into the oceanic lithosphere.

[1] Department of Earth Sciences, Institute of Geophysics, ETH Zürich, Zürich, Switzerland. ✉email: jessica.munch@erdw.ethz.ch

The evolution of boundaries in the Earth's plates mosaic remains one of the key contemporary research questions[1–3]. It has recently been shown that the emergence of modern-style subduction and its catalysts are closely tied to geological processes at the surface of the Earth[1] and that Archean subduction was likely of predominantly retreating character[1,2,4]. Modern retreating intra-oceanic subduction zones can efficiently consume ocean floor and bring material into the mantle by cutting their ways through lithospheric lid interiors (e.g. Scotia, Gibraltar and Caribbean subduction zones, all of which are or have been migrating into the Atlantic[5–14]). Slab retreat is enabled by fractures propagating at the edges of the slab[15], at the so-called STEP-faults (Subduction-Transform Edge Propagator)[16,17]. The fracture trajectories and dynamics are thus crucial to determine how the slab migrates. Different scenarios can be envisaged (Fig. 1): STEP-faults could either exhibit divergent fracturing paths, leading to the consumption of increasingly large portions of the oceanic floor; or parallel fracturing paths, consuming a band of oceanic floor of constant-width; or convergent fracturing paths, resulting in a narrowing of the retreating slab until its detachment. The aim of this study is to investigate which of these scenarios is typical on present-day Earth and if the dominant scenario of lid recycling was different back in geological times.

Fault propagation theory identifies three modes of fracturing[18]: opening (mode I), sliding (mode II) and out-of-plane tearing (mode III). The STEP-fault migration will be strongly controlled by the predominant fracturing mode at the slab edges, in turn controlling the fault migration characteristics and the resulting lid recycling pattern (narrowing, stable or widening). To determine which fracturing mode is favoured in nature and what is the resulting slab retreat trajectory, we performed 3D high-resolution visco-plastic thermomechanical numerical simulations[19] of spontaneously retreating oceanic subduction. Our models start with subduction initiation (Methods, Supplementary Fig. 1a, b) and evolve towards self-sustained slab retreat (Fig. 2).

Our numerical results suggest that both the deformation mechanism and the direction of STEP faults propagation are mainly controlled by the intensity of strain-induced weakening (SIW) of fractures forming at the edges of the slab. Intense SIW causes predominantly brittle STEP-fault propagation associated with the narrowing of the slab until its detachment. In contrast, without such fractures weakening, a preponderantly ductile slab edges propagation occurs that causes slab widening. Comparison of numerical results to existing natural examples suggests that in nature, STEP-faults follow convergent paths that could be controlled by intense fractures weakening due to deep water penetration into the relatively cold oceanic lithosphere of the present-day Earth.

## Results

**General slab evolution in oceanic domains.** Our numerical experiments systematically demonstrate that STEP-faults can spontaneously propagate into the oceanic plate interior starting from pre-existing lithospheric-scale perturbations (Fig. 2, Supplementary Table 2, Supplementary movie 1). The geometries of tearing paths are mainly controlled by the intensity of SIW of brittle/plastic fractures forming inside the lithosphere (Methods). For models with an intense SIW, narrowing slabs (Fig. 2a) and converging tear paths (Fig. 2c, Supplementary Fig. 2a) controlled by out-of-plane (mode III) lithospheric-scale shear are characteristic. In these models, the slab eventually gets so narrow (~120–150 km) that it breaks off and subduction ceases (Supplementary Fig. 2a). In contrast, widening slabs (Fig. 2b) and diverging tear paths (Fig. 2d, Supplementary Fig. 2b) controlled by the combination of out-of-plane shear (mode III) and extensional necking (mode I) of the lithosphere develop in models without SIW. In these models, the slab retreat slows with time (Supplementary Fig. 2b), but subduction typically remains stable and does not stop within the model domain.

**Associated topographic signal.** The two identified retreating subduction scenarios develop distinct topographies (Fig. 2e, f). Models with SIW (Fig. 2a, c, e) result in the formation of a narrow curved arc with a relatively shallow (<8 km deep) trench, similar to what is observed in the Scotia region (cf. Figs. 1 and 2e). We also observe the formation of linear topographic highs on the overriding plate (Fig. 2e, Supplementary Fig. 4) that are parallel to the STEP faults similarly to what is observed in nature (e.g. island chains in Fig. 1). In contrast, models without SIW (Fig. 2f) show the development of a wide straight arc with a deep (>9 km) trench that looks unlike the Scotia subduction zone or other modern narrow retreating subduction zones examples (cf. Figs. 1 and 2f).

**Retreating slabs along continental margins.** We further tested how the two retreating subduction scenarios can be affected by the presence of continental margins. Continental margins are found in proximity to the Caribbean, Scotia (Fig. 1) and also Gibraltar subduction zones, and represent lithospheric-scale heterogeneities[20–22]. These heterogeneities may potentially affect the plate tearing paths and deformation mechanisms. In particular, in the Caribbean, the Lesser Antilles slab has retreated towards the ocean between two continental blocks[23,24] (North and South America) with diverging continental margins. If the slab were to retreat further, STEP-faults could potentially follow these margins, thereby allowing the Caribbean subduction to

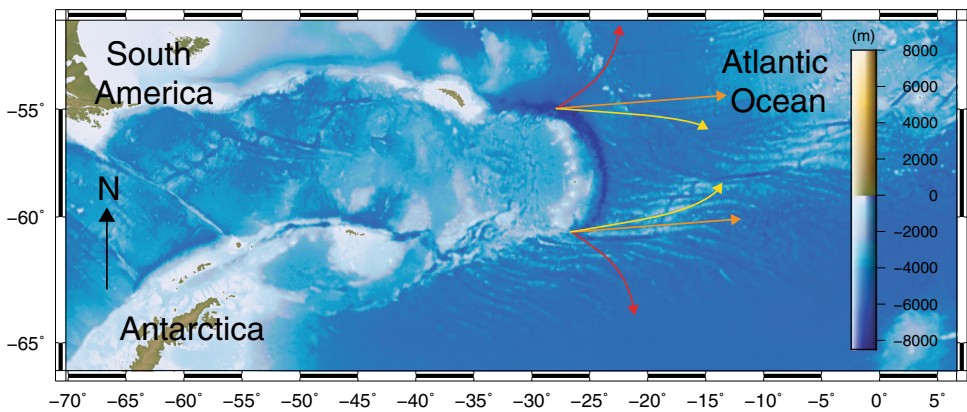

**Fig. 1 Potential migration paths illustrated for the Scotia subduction zone.** Diverging fracturing (red arrows) in the oceanic domain, equidistant fracturing (orange arrows) and converging fracturing (yellow arrows).

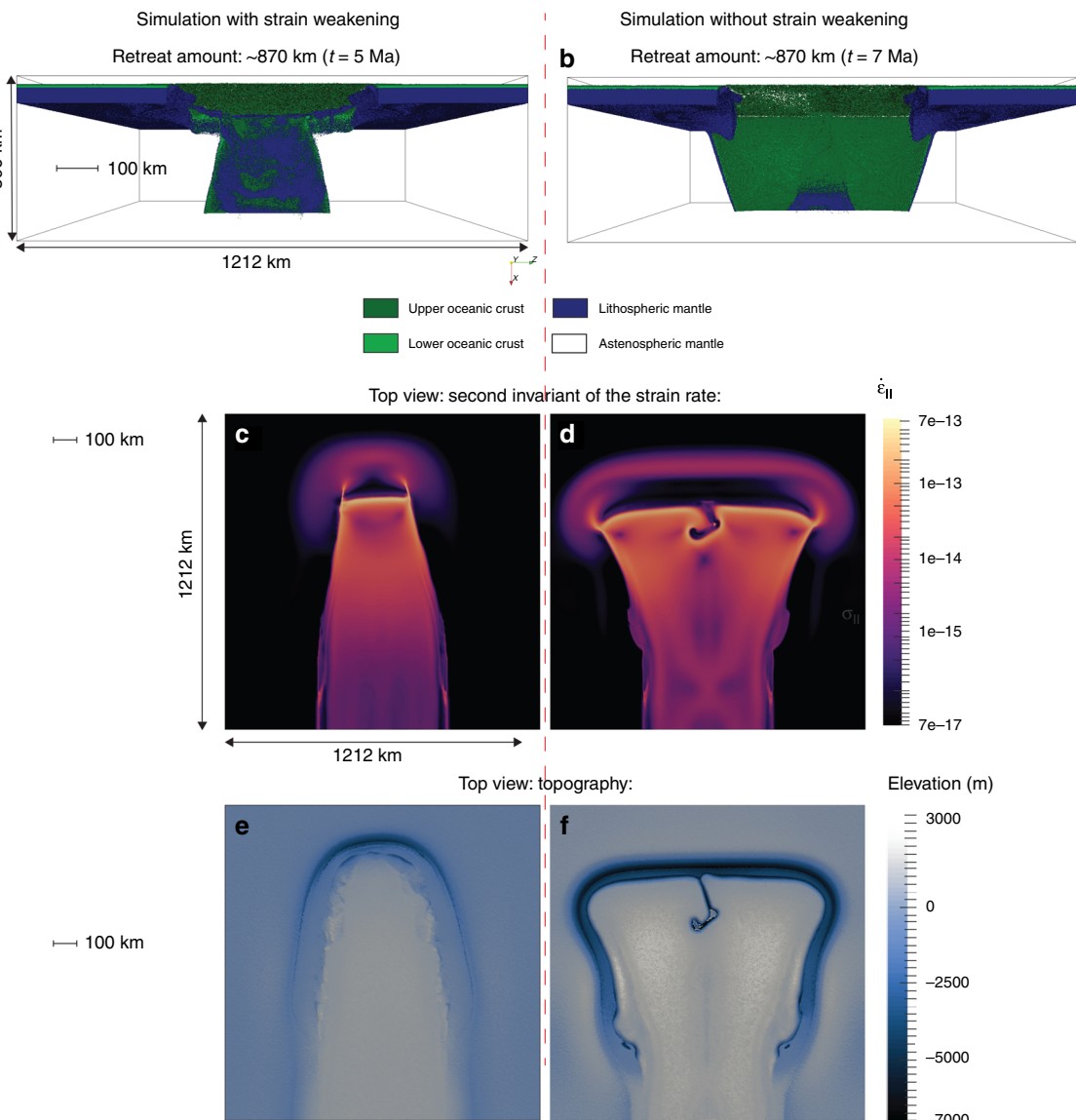

**Fig. 2 Slabs morphology, horizontal deformation and topography with and without strain-induced weakening. a**, **b**: Front view of the retreating slab in the simulation with (**a**) (model swa in Supplementary Table 2) and without (**b**) (model nswa in Supplementary Table 2) strain-induced weakening (SIW) of brittle fractures. **c**, **d** The second invariant of the strain rate at 60 km depth for the simulation with (**c**) and without (**d**) SIW. Both models are shown after ~ 670 km of retreat, ~5 Ma (**a**, **c**, with SIW) and ~7 Ma (**b**, **d**, without SIW) after subduction initiation. Localized regions of increased strain rate are observed at the STEP-faults, which control propagation of tears. These regions are more diffuse in the model without SIW (**b**, **d**). **e**, **f**: Topography of the retreating subduction zones for the models with (**e**) and without (**f**) SIW shown in **a**, **c**, and **b**, **d**, respectively. Note the similarity of topography in **e** to the Scotia region (Fig. 1).

widen and invade the Atlantic[5,25]. We investigated the potential for this future scenario with a simplified oceanic-continental initial plate configuration in which the slab retreats between two continents with equidistant non-weakened (i.e., without any pre-existing weak zones) passive margins in the area where subduction is initiating, the distance between the margins increasing in the direction of slab retreat (Methods, Supplementary Fig. 1b).

Characteristic STEP-fault trajectories are not critically affected by the presence of diverging continental margins both for cases with (Fig. 3a, c) and without (Fig. 3b, d) SIW: the slab retreat trajectories are similar to what we observed in the full oceanic domain and the continental margins are not followed by the tearing paths (Fig. 3) unless pre-existing rheologically weak zones are present along the margins[8,9]. This affirmation is valid both for faster and slower retreating slabs (Supplementary Fig. 3) and confirms what was suggested by Nijholt and Govers[17]. We notice however that the slab narrowing occurs at a

lower rate when the slab retreat is slower and the overriding plate becomes respectively thicker (Supplementary Figs. 2 and 3, and Table 2, slower slabs are resulting from a higher mantle activation volume.).

SIW of brittle/plastic fractures (Methods section) has a strong influence on the subduction zone's retreat since it changes the predominant rheological mechanism of the lithospheric-scale deformation. An intense SIW favours deep penetration of brittle/plastic faults into the lithosphere, whereas absence of the weakening increases the relative influence of ductile lithospheric deformation mechanisms (Methods section).

**Comparison with natural examples.** Simulations with SIW result in topography and retreat patterns similar to what is observed in the Caribbean or Scotia. Indeed, Scotia trench depth is similar to the one we observe in our models (~8 km deep, Fig. 2e)[26–28]. The

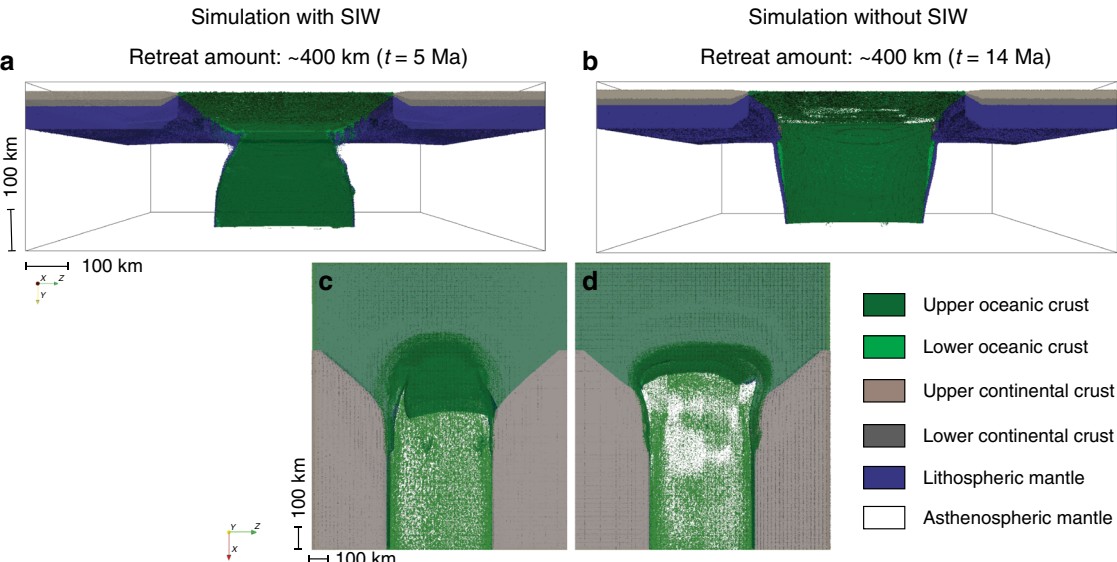

**Fig. 3 Slabs morphology when retreating between two continents with and without strain-induced weakening.** Front (**a**, **b**) and top (**c**, **d**) views of the retreating slabs in the oceanic-continental lithosphere simulations with (**a**, **c**) (model v2swc in Supplementary Table 2) and without (**b**, **d**) (model v2nswc in Supplementary Table 2) strain-induced weakening. The models are shown after ~ 600 km of retreat. Time for the simulation with strain weakening ~5 Ma (**a**, **c**); time for the simulation without strain weakening: ~14 Ma (**b**, **d**).

topographic highs we notice in the track of the subducting slab can also be compared to the structures described in the North and South Scotia ridges[12,26]. Our models for a full oceanic domain show that the STEP-faults are best defined at shallow depth, between 60 and 80 km below the surface. This observation is in accordance with the focal depth of earthquakes measured at the edges of the South Sandwich trench[26,29]. In the Caribbean, seismic data analysis has been used to evidence a detachment of the oceanic Caribbean plate from the continental South American plate via STEP-fault propagation[30], which is coherent with our model results. Moreover, a narrowing slab trajectory has been suggested by studies using wide angle seismics[14].

All of our experiments involving SIW show a narrowing of the slab due to convergent STEP-faults, including when the lithosphere gets younger in the direction of retreat (Supplementary Fig. 4). The slower the retreat the more equidistantly the STEP faults propagate.

It is however hard to assess the exact influence of the South Sandwich fracture zone or small continental blocks around the Caribbean or Scotia (e.g. South Georgia) on the slab retreat. From our modelling results, the tendency of a retreating subduction zone to follow a fracture zone greatly depends on the rheology of the fracture zone: large weak discontinuities can be followed by STEP-faults[8,16] whereas small and less weak structures will be predominantly ignored.

## Discussion

The impact of SIW intensity has strong implications for both modern and Archean Earth, and other terrestrial planets (e.g. Venus). Modern Earth is thought to have a colder and thus more rigid crust and mantle than Archean Earth[31–33]. Colder conditions favour brittle deformation[34] and deep water penetration along faults[8,20], causing the development of weak hydrous minerals in the deforming lithosphere[35,36], thereby creating favourable conditions for an intense SIW (Methods) and converging lid consumption patterns observed today, but this may not be valid during the onset of plate tectonics in the Archean.

Spontaneously retreating subduction zones that cut through the plate interior do not necessarily require modern-style plate

tectonics. They could also develop within a single/stagnant lid as the result of other (e.g. plume-induced[2,37,38] and/or meteorite impact-induced[39,40]) processes. This peculiar type of subduction may therefore control partial or complete recycling of the lid, including global or regional overturn events, on early Earth and other terrestrial planets[2,38,40,41]. If, on Modern Earth, converging STEP-faults and narrowing slabs seem to be favoured; with warmer and/or dry lithospheric conditions, where there is no significant brittle/plastic strain weakening due to limited or absent hydration along fractures, such as on early Earth or on Venus[42], the simulations without SIW which favour ductile lithospheric deformation may be more representative. There, the retreating subduction zones should widen with time, ultimately leading to global lid recycling as suggested for Venus[42]. On early Earth, upon gradual cooling of the mantle, the propagation of STEP-faults during episodic retreating subduction may also have enabled the creation of new plate boundaries and contributed to the gradual emergence of a global plate mosaic that is crucial for establishing modern-style plate tectonics[42].

## Methods

**Numerical approach.** We model subduction initiation and slab retreat using the 3D thermo-mechanical code I3ELVIS[2].

Mass, momentum and energy conservation are solved on a non-deforming Eulerian staggered grid and the physical properties of rocks (temperature, rock type, etc.) are advected via Lagrangian markers. Detailed code description is given in[19].

**Model rheology.** The code accounts for visco-plastic deformation using an effective viscosity formulation for the material. The ductile deformation comprises diffusion and dislocation creep as described in Gerya et al.[2]. The ductile creep viscosity $\eta_{\text{ductile}}$ is calculated taking into account dislocation and diffusion creep as:

$$\frac{1}{\eta_{\text{ductile}}} = \frac{1}{\eta_{\text{diff}}} + \frac{1}{\eta_{\text{disl}}}$$

where $\eta_{\text{diff}}$, the effective viscosity for diffusion creep, is computed as

$$\eta_{\text{diff}} = \frac{A_{\text{D}}}{2\sigma_{\text{cr}}^{n-1}} \exp\left(\frac{E + PV}{RT}\right)$$

and $\eta_{\text{disl}}$, the effective viscosity for dislocation creep is computed as

$$\eta_{\text{disl}} = \frac{A_{\text{D}}^{\frac{1}{n}}}{2} \exp\left(\frac{E + PV}{nRT}\right) \dot{\varepsilon}_{II}^{\frac{1}{n}-1}$$

where $A_D$, $E$, $V$ and $n$ are respectively the pre-exponential factor, the activation energy and the activation volume; they are experimentally determined constants. $R$ is the noble gas constant, $P$ is the pressure, $T$ the temperature, $\dot{\varepsilon}_{II}$ the second invariant of the deviatoric strain rate tensor and $\sigma_{cr}$ the diffusion-dislocation transition stress (Supplementary Table 1). Partial melting, melt extraction and melt-induced weakening are modeled following Gerya et al.[2].

Rheological layers are defined as follows (Supplementary Table 1). The ductile rheology of the oceanic crust is defined using the wet quartzite (upper crust) and anorthosite (lower crust) flow laws[43], the continental crust ductile rheology is defined using the wet quartzite flow law[43], the lithospheric and asthenospheric mantle rheology is defined based on dry olivine flow law[43] whereas the shear zone material is assigned a wet olivine rheology[43]. The boundary between the lithospheric and asthenospheric mantle is purely thermal and is set at 1273 K.

The ductile rheology is combined with brittle/plastic rheology based on a Drucker-Prager yielding criterion:

$$\eta_{\text{ductile}} \leq \frac{C_{\text{eff}} + \gamma_{\text{eff}} P \lambda_{\text{melt}}}{2\dot{\varepsilon}_{II}},$$

$$\lambda_{\text{melt}} = 1 - \frac{P_{\text{melt}}}{P}$$

where $C_{\text{eff}}$ is the material cohesion, $\gamma_{\text{eff}}$ is effective friction coefficient, $\lambda_{\text{melt}}$ the long-term melt-induced weakening factor[2], $\dot{\varepsilon}_{II}$ is the second invariant of the strain rate tensor, $P_{\text{melt}}$ is the melt pressure and $P$ is the total pressure.

It should be noted that our rheological model neglects the elastic deformation. As the dominant large strain associated with the STEP faults propagation is visco-plastic, this simplification should not affect the model behaviour in a critical manner.

**Strain-induced weakening.** One of the critical factors in our numerical experiments is the SIW of brittle fractures forming in the lithosphere[35,44]. This weakening is related, in particular, to deep water percolation along deforming oceanic fault zones leading to their intense hydration[32,35] which strongly decreases the strength of fractured fault rocks[35,36,45]. In our numerical model, we use a simplified strain weakening model that prescribes a linear decrease of the internal friction coefficient with increasing cumulative brittle/plastic strain $\varepsilon_c$[44,46].

$$\gamma_{\text{eff}} = b_0$$

when $\varepsilon_c < \varepsilon_1$

$$\gamma_{\text{eff}} = b_0 + (b_1 - b_0)\frac{\varepsilon_c - \varepsilon_1}{\varepsilon_2 - \varepsilon_1}$$

when $\varepsilon_1 < \varepsilon_c < \varepsilon_2$

$$\gamma_{\text{eff}} = b_1$$

when $\varepsilon_c > \varepsilon_2$

$$\varepsilon_c = \int \sqrt{\frac{1}{2}\left(\dot{\varepsilon}_{\text{ij(plastic)}}\right)^2} \, dt$$

Where $\gamma_{\text{eff}}$ is the internal friction coefficient, $\varepsilon_1$ and $\varepsilon_2$ are two threshold strains, $b_0$ and $b_1$ are the initial and final internal friction coefficient values, $\dot{\varepsilon}_{\text{ij(plastic)}}$ is the brittle/plastic strain rate tensor[47].

**Numerical model setup.** We model two different settings using regional-scale models: a purely oceanic domain and a mixed oceanic/continental domain. The models are $1212 \times 1212$ km wide in both horizontal directions and 392 km deep. We use a grid resolution of 2 and 3 km in the vertical and horizontal directions, respectively, which is sufficient to resolve plate boundaries and the lithological and rheological structures of both oceanic and continental domains. In both cases, we model slab evolution from subduction initiation to slab retreat. We initiate subduction using a strong lateral lithospheric age contrast (and hence density contrast) in the oceanic lithosphere. The lateral boundaries of the box are mechanically defined as free slip conditions, and no heat flux across these boundaries is allowed. The top boundary is free slip, and fixed temperature. No external forcing is applied, and the model is evolving entirely due to its internal dynamics. A new upper crust is gradually forming by cooling along with slab retreat. In addition, the uppermost part of the model interior (20 km) is composed of a sticky air layer[38], which is a low density, low viscosity layer that emulates a free surface condition on top of the lithosphere. The lower boundary of the model is set as an open boundary both mechanically and for the temperature. Temperature is set at 273 K at the surface of the lithosphere and in the sticky air layer. The temperature in the lithosphere is calculated according to a cooling half-space model, where we set an initial age of 40 Ma for the lithosphere at the beginning of the simulation. In the asthenosphere, a positive gradient of 0.5 K/km is defined. Our limited model domain - which is virtually extended by the open boundary – thus does not implicitly include the 660-km upper-lower mantle transition that can potentially affect the retreating subduction dynamics, but is not subjected to undesirable interaction with a shallow (392 km) model boundary either. Since in most of our numerical experiments the modeled amount of slab subduction is <600 km this simplification is an approvable

compromise. Due to the computational restrictions, our study focuses on high-resolution 3D modeling of relatively narrow retreating subduction zones limited by STEP faults that evolve in the uppermost 660 km.

We investigated numerically two model settings (with and without continents) shown in Supplementary Fig. 1a, b. In both model setups, we start the simulation from a homogeneous 40-Ma old lithosphere throughout, with the exception of a rectangular $200 \times 460$ km domain of younger lithospheric age at the left model boundary (1–10 ka, Supplementary Fig. 1b, circled in red). No slab is prescribed. In this "lithospheric window" the cooling age of the lithosphere decreases from 10 to 1 ka from the border of the box towards the area where we want to trigger slab formation (the 1-ka part of the young window being located next to the area where the slab forms, Supplementary Fig. 1a, b). In order to enable the retreating subduction initiation, we prescribe a "H" shaped weak ($\gamma_{\text{eff}} = 0$) shear zone pattern at the lithospheric window that extends by ~200 km into the old lithosphere (Supplementary Fig. 1a). This highly simplified model geometry is used to prescribe both the initial slab width and the direction of free subduction retreat without a need to impede internal dynamics by forcing convergence along the boundaries, nor setting an initial slab dip, the model is hence fully dynamic.

In the mixed oceanic/continental simulation, we added two continental margins at the lateral model boundaries (Supplementary Fig. 1b). These are equidistant around the area where the slab forms and the distance between the margins increases in the direction of slab retreat. The continents are located ~500 km apart of each other and their margins stay parallel for ~ 400 km (Supplementary Fig. 1b).

## Data availability

Due to the very large size of the data, the model results are available from the corresponding author upon request.

## Code availability

Researchers interested in using I3ELVIS code should contact T. V. Gerya (taras.gerya@erdw.ethz.ch).

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

## Acknowledgements

This study was funded by the European Union's Horizon 2020 research and innovation program SUBITOP under the Sklodowska-Curie grant agreement No 674899 and by the SNF research grant 200021_182069. Suggestions and comments by Suzanne Atkins, Marie Bocher, Casper Pranger and Martina Ulvrova are appreciated. All simulations were performed on the ETH-Zurich Euler and Leonhard clusters. Scotia region map was made using GMT software and data from NOAA. The open-source software ParaView (http://www.paraview.org) was used for 3D visualizations of the experiments. Oslo colorscale from http://www.fabiocrameri.ch was used for the topography. Annotations on figures were made using Adobe Illustrator software.

## Author contributions

J.M. designed the study, conducted the numerical experiments and interpreted the results. T.V.G. designed the study and the 3D thermomechanical code. K.U. designed the study and interpreted the results. All authors discussed the results, problems and methods, interpreted the data and wrote the paper.

## Competing interests

The authors declare no competing interests.
