## [Peer Review File · Nature Communications]

Reviewers' comments:

Reviewer #1 (Remarks to the Author):

Report on the paper: "Lid recycling controlled by slab edge weakening" by J. Munch et al.

The present paper employs 3D geodynamic numerical modelling to investigate the fundamental dynamics governing the evolution of slab roll-back intra-oceanic subduction zones, referring to natural examples such as the Caribbean, South Scotia and Gibraltar (Atlantic) Arcs.

Numerical modelling is used to test the importance of two main parameters in the propagation of so-called STEP faults (Subduction-Transform Edge Propagator) that assist the subduction retreat migration, these are: a) the predominant rheological mechanism of the implied deformation (i.e. existence vs. absence of strain-induced weakening - SIW); and b) the existence vs. absence of bounding passive continental margins (as to see if the geometry of these margins could drive the lateral propagation of the subduction zone).

The propagation direction of STEP faults (either complying with a convergent or a divergent geometry) is critical, since it would ultimately determine whether this type of subduction zone is prone to progressively narrow, leading to the ("necking") break-off of the subducting slab, or alternatively to the widening "expand" of this subduction along the circum-Atlantic passive margins.

The 4 main conclusions provided by the paper are:

- 1) The main factor governing the geometry of propagation of STEP faults is strain induced weakening (SIW): this weakening renders STEP fault convergence, whether its absence leads to the widening, lateral propagation, of the subduction zone.
- 2) The existence of passive continental margins laterally bounding the initial subduction-arc bears no relevant influence on the propagation geometry of STEP faults and ensuing subduction-arc evolution.
- 3) Present-day subduction arc geometry (e.g. Atlantic South Scotia and Caribbean arcs) is preferentially compatible with fault-controlled fluid-driven chemical weakening (promoting SIW), along more discrete (localized) STEP domains, as would be expected to occur in a colder stronger lithospheric lid (such as the presently existent one).
- 4) Given the observed compatibility with Present-day natural examples, Atlantic subduction arcs are not expected to evolve in a way as to propagate this subduction along its margins (and thus to invade or "infect" the Atlantic), but rather to undergo future ("necking") slab break-off. Conversely, a relatively hotter and dryer lithospheric lid (preventing or strongly hampering SIW) in the Archean could have favoured a scenario of widening lateral propagation subduction evolution in that geological past.

I find this paper extremely important. It is a new, well-argued, contribution to understand the dynamics (the main rheological mechanism) assisting STEP fault propagation, and thus, ultimately, the tectonic evolution of isolated subduction arcs within major oceanic basins.

This bears the huge implication of trying to know if these arcs could correspond to the initial locus of (lateral) propagation of subduction zones (along the pristine passive margins of the encompassing oceanic basin), potentially leading to the closure of a main ocean. As such, it has implications for the understanding of the evolution of Wilson and Supercontinent Cycles, as well as for the knowledge regarding the workings of Plate Tectonics through geological time (e.g.

regarding the issue of trying to know whether the presently observed plate configuration was set at a relatively early stage of Earth evolution, and has since then acquired a steady-state configuration, or if, on the contrary, such a configuration is prone to continuous change and evolution).

The (3D numerical modelling) methodology is supported by well-designed models adequately conceived to address the two main investigated parameters. Noticeably, these models are fully dynamical, i.e. the forces driving the resulting geometry/kinematics are not externally imposed to the model, but rather are exclusively the result of the buoyancy contrast and viscous resistance between the different prescribed model domains/units.

The paper is well written, well-illustrated, and cites the most relevant previous contributions.

As such, in my opinion, the present work will be of interest for a broad audience of researchers and (even) for Earth-Science educated non-specialists. It absolutely merits publication (as soon as possible) in Nature Communications.

I have only two main points that I think the authors should consider:

1. I do not fully agree with the assertion that the study subduction arcs cannot evolve as to propagate laterally, sub-parallel to the passive margins of the confining main ocean, ultimately triggering the future closure of the latter. In fact, the investigated parameters are not the only ones governing the tectonic evolution of these arcs. Noticeably, this type of arcs in the Atlantic are migrating (retreating) towards the inner domains of the main ocean, i.e. towards the middle oceanic ridge. As such, the outer convex front of the arcuate subduction zone is progressively encountering younger, hotter and more buoyant oceanic lithosphere, which is less prone to be subducted. Contrastingly, at the extremities of the arc, the subducting oceanic lithosphere is further away from the oceanic ridge, and thus is older (i.e. relatively colder and denser) and hence more prone to undergo subduction. Note that these arc extremities correspond to the domains where STEP faults have to propagate to assist arc retreat. Nobody knows what would happen, or what will happen, when (e.g.) the Scotia Arc will get closer to the mid oceanic ridge, but it is conceivable that as increasingly younger oceanic lithosphere (at its convex front) becomes harder to subduct the arc would widen (open?) through lateral propagation of the subduction zone. If this process of formation of widen intra-oceanic subduction is possible, then it could also be nucleated at the other two Atlantic oceanic arcs (Caribbean and Gibraltar), potentially creating a network of extended subduction zones, and allowing to speculate in favor of the future closure of the Atlantic. In the present models, outside the younger window of overriding lithosphere, all (potentially) subducting lithosphere has the same original age (40 My) all around/near the arc trench. This prevents investigating the dynamic impact of having younger subducting lithosphere at the arc front, and older at its extremities, in the evolving arc retreat migration (including possible effects on the propagation of the STEP faults that assist such retreat). I agree that such an approach would be out of the scope of the present work (it would correspond to a new paper by itself alone), but I think this matter would merit some brief discussion while addressing the limitations of the present models.

2. In their models the authors consider a maximum basal depth of only ~390 km (to which an open bottom boundary condition is prescribed), thus not including the upper-lower mantle transition at 660 km. This transition has been known to influence the dynamics of the subducting slab, specifically regarding the evolving shift between slab roll forward and slab roll-back, and the achievement of a trench-retreat steady-state (e.g. Stegman et al., 2010). As such, the interaction between the slab and the 660 km transition could also be meaningful for the dynamics of surface arc-propagation, namely since the bending of the retreating slab due to mantle toroidal flow would interfere/interact with the bending caused by the arrival of the subducting slab to the upper-lower

mantle transition. This limitation of the models (i.e. absence of 660 km upper-lower mantle transition) would perhaps also merit some brief discussion.

Some other minor points regarding the Methods section (besides very few other directly annotated throughout the original ms):

a) Please clearly state what is the original age of the prescribed window of younger overriding lithosphere.

b) Clearly state that the model is fully dynamic (only driven by internal buoyancy contrast and viscous resistance forces), not subjected to any external forces.

c) Model original state: please clarify if there is an initial portion of the slab prescribed to be already subducted in the experimental initial state. If so, please specify the angle and maximum depth of such initial state subducting slab.

Filipe M. Rosas
IDL, University of Lisbon

References cited:

Stegman, D. R., Schellart, W. P. & Freeman, J. A., 2010. Competing influences of plate width and far-field boundary conditions on trench migration and morphology of subducted slabs in the upper mantle. *Tectonophysics*, 483 (1-2): 46-57. doi.org/10.1016/j.tecto.2009.08.026

Reviewer #2 (Remarks to the Author):

The presented work is original and innovative: it explores the role of strain weakening in the development of slab tear propagation near STEP-faults. The numerical models are of the highest scientific quality. The paper is very well-written. The ideas in the paper regarding the evolution of subduction during Earth history are thought-provoking but, as detailed further below, insufficiently substantiated as yet because the connection between the models and particularly present-day STEP-bounded subduction systems is poorly developed.

The strain weakening law is a critical ingredient in the numerical experiments. It refers to an aspect of rock material behavior that is commonly used in numerical modeling studies to induce localization. It is not directly constrained by independent observations, e.g., from rock mechanical experiments. Given the uncertainty surrounding the strain weakening parameters, the authors understandably vary them. They find substantial and interesting variations in the predicted tearing (diverging, straight, or converging).

In line 46-7 it is written that "The aim 46 of this study is to investigate which of these scenarios is typical on present-day Earth". What is conspicuously missing in the study is a comparison to present-day observations, which would allow the authors to conclude whether strong or weak strain weakening dominates at a dozen or so STEP-faults. An onset towards a comparison is in sentence 80-82: "models without SIW show the development of a wide straight arc with a deep (>9 km) trench that looks unlike the Scotia subduction zone or other modern narrow retreating subduction zones examples (cf. Figs. 1 and 2.f.)". The correspondence between Figs. 1 and 2f is not so apparent to the reader as suggested here. As the emphasis of the paper is on Gibraltar, Scotia and Caribbean STEPs, recent geological indicators of STEP convergence need to be discussed here and a semi-quantitative comparison to the model predictions needs to be made. Wide subduction systems like Tonga and New Hebrides do not seem to show STEP-convergence, making me wonder whether the model experiments are valid only for rather narrow trenches. In

terms of the paper structure, benchmarking these critical weakening parameters is a logical step before backstrapolating to historical settings, and before speculating about the emergence of plate tectonics.

Spakman and Wortel defined the “STEP Fault” as the strike-slip boundary between the two surface plates, one of which is the overriding plate. This definition appears to be widely followed in the literature. The “free subduction” numerical models in the paper do not have an overriding plate so there are no STEP-faults in these models, but tears.

Could the prediction that the present Atlantic subduction zones will cease to exist be affected by an overriding plate? Please discuss particularly the potential sensitivity of mode III fracturing to this; my first thought would be that out-of-plane also will require deformation of STEP-fault adjacent parts of the overriding plate, and that mechanical resistance to such deformation may affect the tearing in this coupled system.

Opening sentences (L.33-37). “The puzzling emergence of plate tectonics and the maintenance of a mosaic of coherent 33 plates remains one of the key contemporary research questions”; this is absolutely true, but the contribution of the current research to answering this question is very limited.

Figure 1. The southern end of the Scotia STEP may actually not be tearing because further to the east there already is a plate boundary discontinuity. I looked it up in Govers and Wortel (2005), and on page 510 they say: “Beyond the STEP further to the east, the South America and Antarctica plates are direct neighbors along an east–west oriented transform plate boundary (South America–Antarctica (SA- A) ridge). This has the consequence that STEP fault lengthening here occurs along a pre-existing plate boundary.”

In Line 60, add “free-subduction” as an adjective to “models”, this clarifies an important aspect of the approach.

Line 118, specify what “critically” means.

L.118. This confirms the conclusions of Nijholt, N., & Govers, R. (2015). The role of passive margins on the evolution of Subduction-Transform Edge Propagators (STEPs). *Journal of Geophysical Research*, 120(10), 7203–7230. It would be fair to credit these authors here.

L. 163: State explicitly and motivate that rock elasticity is neglected. Such rigid-visco-plastic rheology differs from actual bulk rock properties, but is it a relevant difference?

Sentence L.213-4: “We use a grid resolution of 2 and 3 km in the vertical and horizontal directions, respectively.” There is no motivation of these choices and the dependence of the results on the grid size and time step which would be appropriate in this methods section.

L.239 The authors assume the same water content for oceans and continents. There is a substantial literature starting from Hirth and Kohlsted suggesting that oceanic lithosphere initially is much drier.

Reviewer #1 (Remarks to the Author):

1. I do not fully agree with the assertion that the study subduction arcs cannot evolve as to propagate laterally, sub-parallel to the passive margins of the confining main ocean, ultimately triggering the future closure of the latter. In fact, the investigated parameters are not the only ones governing the tectonic evolution of these arcs. Noticeably, this type of arcs in the Atlantic are migrating (retreating) towards the inner domains of the main ocean, i.e. towards the middle oceanic ridge. As such, the outer convex front of the arcuate subduction zone is progressively encountering younger, hotter and more buoyant oceanic lithosphere, which is less prone to be subducted. Contrastingly, at the extremities of the arc, the subducting oceanic lithosphere is further away from the oceanic ridge, and thus is older (i.e. relatively colder and denser) and hence more prone to undergo subduction. Note that these arc extremities correspond to the domains where STEP faults have to propagate to assist arc retreat. Nobody knows what would happen, or what will happen, when (e.g.) the Scotia Arc will get closer to the mid oceanic ridge, but it is conceivable that as increasingly younger oceanic lithosphere (at its convex front) becomes harder to subduct the arc would widen (open?) through lateral propagation of the subduction zone. If this process of formation of widen intra-oceanic subduction is possible, then it could also be nucleated at the other two Atlantic oceanic arcs (Caribbean and Gibraltar), potentially creating a network of extended subduction zones, and allowing to speculate in favor of the future closure of the Atlantic. In the present models, outside the younger window of overriding lithosphere, all (potentially) subducting lithosphere has the same original age (40 My) all around/near the arc trench. This prevents investigating the dynamic impact of having younger subducting lithosphere at the arc front, and older at its extremities, in the evolving arc retreat migration (including possible effects on the propagation of the STEP faults that assist such retreat). I agree that such an approach would be out of the scope of the present work (it would correspond to a new paper by itself alone), but I think this matter would merit some brief discussion while addressing the limitations of the present models.

Discussion on passive margins has been modified. One possible reason for passive margin activation could be existence of a weak zone along the margin (e.g., Spackman et al., 2018) that is not modeled in our numerical experiments. We also performed an additional simulation where the lithosphere gets younger in the direction of slab retreat, in a simulation involving SIW, which would mimic the slab propagation in the direction of a middle oceanic ridge. Our simulation results in the same main observation as when there is no age transition in the lithosphere: the slab narrows (see extended data figure 4). The slab behavior is however not exactly the same. It appears to detach much earlier than in the case without age transition in the lithosphere. A more detailed study of this point is indeed required but would be more adapted in a paper in itself.

2. In their models the authors consider a maximum basal depth of only ~390 km (to which an open bottom boundary condition is prescribed), thus not including the upper-lower mantle transition at 660 km. This transition has been known to influence the dynamics of the subducting slab, specifically regarding the evolving shift between slab roll forward and slab roll-back, and the achievement of a trench-retreat steady-state (e.g. Stegman et al., 2010). As such, the interaction between the slab and the 660 km transition could also be meaningful for the dynamics of surface arc-propagation, namely since the bending of the retreating slab due to mantle toroidal flow would interfere/interact with the bending caused by the arrival of the subducting slab to the upper-lower mantle transition. This limitation of the models (i.e. absence of 660 km upper-lower mantle transition) would perhaps also merit some brief discussion.

We agree that the upper-lower mantle transition can affect the retreating subduction dynamics. However in most of our numerical experiments, the amount of modeled slab retreat is less than 600 km, which means that the modelled slabs are not reaching the 660 km transition. This is the reason why we have not taken this boundary into account in our study. The explanation is added to the model description.

Some other minor points regarding the Methods section (besides very few other directly annotated throughout the original ms):

We adapted the method section to your suggestions.

a) Please clearly state what is the original age of the prescribed window of younger overriding lithosphere.

We emphasized the age of the lithospheric window in the manuscript. The age structure of the lithospheric patch is available in line 282++.

b) Clearly state that the model is fully dynamic (only driven by internal buoyancy contrast and viscous resistance forces), not subjected to any external forces.

We added a sentence to state that there is no external forcing.

c) Model original state: please clarify if there an initial portion of the slab prescribed to be already subducted in the experimental initial state. If so, please specify the angle and maximum depth of such initial state subducting slab.

We added an explicit statement that there is no initial slab.

Reviewer #2 (Remarks to the Author):

In line 46-7 it is written that “The aim 46 of this study is to investigate which of these scenarios is typical on present-day Earth”. What is conspicuously missing in the study is a comparison to present-day observations, which would allow the authors to conclude whether strong or weak strain weakening dominates at a dozen or so STEP-faults. A onset towards a comparison is in sentence 80-82: “models without SIW show the development of a wide straight arc with a deep (>9 km) trench that looks unlike the Scotia subduction zone or other modern narrow retreating subduction zones examples (cf. Figs. 1 and 2.f.)”. The correspondence between Figs.1 and 2f is not so apparent to the reader as suggested here.

The discussion is clarified. Figure 2f (model WITHOUT strain-induced weakening) is indeed dissimilar to Figure 1 whereas Figure 2e (model WITH strain-induced weakening) shows many common topographic features with Figure 1. This allows us to conclude that without strain weakening, we do not get nature like topographic signals. We limited our comparison to classical cases of narrow subduction zones limited by STEP faults that have relatively simple retreat dynamics and clear expression of topographic signatures. Comparison with other STEP faults is more problematic since propagation dynamics and topographic expression of respective subduction zones is more complex and difficult to interpret and/or the width of subducting slabs is significantly larger than we were able to model at high-resolution.

As the emphasis of the paper is on Gibraltar, Scotia and Caribbean STEPs, recent geological indicators of STEP convergence need to be discussed here and a semi-quantitative comparison to the model predictions needs to be made.

We added a paragraph comparing our models to natural examples.

Wide subduction systems like Tonga and New Hebrides do not seem to show STEP-convergence, making me wonder whether the model experiments are valid only for rather narrow trenches. In terms of the paper structure, benchmarking these critical weakening parameters is a logical step before backstrapolating to historical settings, and before speculating about the emergence of plate tectonics.

Clarification is added to the model description. Our study focuses on high-resolution modeling of narrow retreating subduction zones limited by STEP faults. Due to computational restrictions, to be able to perform our 3D simulation at high resolution, we had to limit lateral size of our computational domain by 1212 km and were thus only able to model subduction zones width up to 660 km. The conclusions we are making regarding to retreating subduction zones are thus only valid for such narrow retreating subduction examples. High-resolution models of wider retreating subduction zones can indeed become feasible in the future.

Spakman and Wortel defined the “STEP Fault” as the strike-slip boundary between the two surface plates, one of which is the overriding plate. This definition appears to be widely followed in the literature. The “free subduction” numerical models in the paper do not have an overriding plate so there are no STEP-faults in these models, but tears.

Due to the finite cooling age in the "young window", there is actually always an overriding plate in our models. It is very thin in the beginning but as the slab retreat, thanks to cooling, the overriding plate get thicker (highlighted in the red box in the figure here below). In the following figure, the blue part is the solidified overriding plate (above a visually dominant, black body a partially molten material). It is clearly highlighted in the viscosity plot below especially in the vicinity of the slab.

Could the prediction that the present Atlantic subduction zones will cease to exist be affected by an overriding plate? Please discuss particularly the potential sensitivity of mode III fracturing to this; my first thought would be that out-of-plane also will require deformation of STEP-fault adjacent parts of the overriding plate, and that mechanical resistance to such deformation may affect the tearing in this coupled system.

We added short sentence on the potential influence of the overriding plate thickness on the STEP faults geometry.

Opening sentences (L.33-37). "The puzzling emergence of plate tectonics and the maintenance of a mosaic of coherent 33 plates remains one of the key contemporary research questions"; this is absolutely true, but the contribution of the current research to answering this question is very limited.

We reorganized the sentence and removed the reference to the emergence of plate tectonics..

Figure 1. The southern end of the Scotia STEP may actually not be tearing because further to the east there already is a plate boundary discontinuity. I looked it up in Govers and Wortel (2005), and on page 510 they say: "Beyond the STEP further to the east, the South America and Antarctica plates are direct neighbors along an east–west oriented transform plate boundary (South America–Antarctica (SA- A) ridge). This has the consequence that STEP fault lengthening here occurs along a pre-existing plate boundary."

We added a sentence that the ability of STEP faults to follow preexisting structures will mainly depend on their rheological properties: large and weak structures can be followed whereas small and less weak structures will be predominantly ignored.

In Line 60, add “free-subduction” as an adjective to “models”, this clarifies an important aspect of the approach.

Commonly, free subduction models (e.g., analogue or numerical models) are rather simplified and contain an isolated single subducting plate embedded into a low-viscosity fluid. Since our models contain an overriding plate (though relatively thin) and spontaneously propagating STEP faults we prefer not to define them as free subduction models.

Line 118, specify what “critically” means.

Specified by adding a half-sentence

L.118. This confirms the conclusions of Nijholt, N., & Govers, R. (2015). The role of passive margins on the evolution of Subduction-Transform Edge Propagators (STEPs). *Journal of Geophysical Research*, 120(10), 7203–7230. It would be fair to credit these authors here.

We added the suggested reference in the manuscript.

L. 163: State explicitly and motivate that rock elasticity is neglected. Such rigid-viscoplastic rheology differs from actual bulk rock properties, but is it a relevant difference?

The potential role of elasticity is discussed in short in the Method section. As the dominant large strain associated with the STEP faults propagation is visco-plastic, neglecting elasticity should not affect the model behavior in a critical manner. This technical limitation is left for future research.

The magnitude of topography, which we discuss in no great detail, may be more sensitive to elasticity, but Bottrill et al. (2012, *Solid Earth*, doi:10.5194/se-3-387-2012) have indicated that the principal features of topography remain unchanged.

A statement to this effect with an explicit mentioning as limitation has been added to the manuscript.

Sentence L.213-4: “We use a grid resolution of 2 and 3 km in the vertical and horizontal directions, respectively.” There is no motivation of these choices and the dependence of the results on the grid size and time step which would be appropriate in this methods section.

The discussion of model resolution has been extended. The choice of this resolution is led by the highest possible resolution keeping computational costs viable. This

resolution also allows to accurately resolve the lithological and rheological structure of both oceanic and continental domains.

L.239 The authors assume the same water content for oceans and continents. There is a substantial literature starting from Hirth and Kohlstedt suggesting that oceanic lithosphere initially is much drier.

We decided to ignore rheological differences between oceanic and continental mantle lithosphere due to uncertainties in the composition and water content for the continental lithosphere. On the one hand, continental lithosphere (especially cratonic roots) is often considered to be dry and melt-depleted (and thus rheologically strong) that explains its longevity (e.g., discussion in Gerya, 2014, Gondwana Research and references therein). On the other hand, continental lithosphere can also become hydrated and metasomatised (and thus rheologically weak).

REVIEWERS' COMMENTS:

Reviewer #1 (Remarks to the Author):

I find the reply to my previous comments perfectly satisfactory. In my view the paper, which I already considered a very interesting contribution, has now improved even further and fully deserves being published in Nature Communications.

Filipe M. Rosas

Reviewer #2 (Remarks to the Author):

The authors have responded to, and modified the manuscript, my comments on the previous version. The manuscript is a balanced representation of the numerical results, and of their implications with one exception; technically they do not use a free subduction approach, but the overriding plate is unrealistically thin and this aspect is not very explicit in the revised manuscript.

REVIEWERS' COMMENTS:

Reviewer #1 (Remarks to the Author):

I find the reply to my previous comments perfectly satisfactory. In my view the paper, which I already considered a very interesting contribution, has now improved even further and fully deserves being published in Nature Communications.

Filipe M. Rosas

Reviewer #2 (Remarks to the Author):

The authors have responded to, and modified the manuscript, my comments on the previous version. The manuscript is a balanced representation of the numerical results, and of their implications with one exception; technically they do not use a free subduction approach, but the overriding plate is unrealistically thin and this aspect is not very explicit in the revised manuscript.

We thank the reviewers for their inputs which were very helpful in improving our study. We added a sentence specifying that the overriding plate is gradually forming by cooling along with slab retreat (l 242-243).